# Role of Peripheral Inflammatory Markers in Patients with Acute Headache Attack to Differentiate between Migraine and Non-Migraine Headache

**DOI:** 10.3390/jcm11216538

**Published:** 2022-11-03

**Authors:** Sang-Hwa Lee, Jong-Ho Kim, Young-Suk Kwon, Jong-Hee Sohn

**Affiliations:** 1Department of Neurology, Chuncheon Sacred Heart Hospital, Hallym University College of Medicine, Chuncheon 24252, Korea; 2Institute of New Frontier Research Team, Hallym University College of Medicine, Chuncheon 24252, Korea; 3Department of Anesthesiology and Pain Medicine, Chuncheon Sacred Heart Hospital, Hallym University College of Medicine, Chuncheon 24252, Korea

**Keywords:** headache, migraine, neutrophil-to-lymphocyte ratio, neutrophil-to-monocyte ratio, inflammation

## Abstract

Although the potential relationship between headaches, particularly migraine, and peripheral inflammatory markers (PIMs) has been investigated, it is unclear whether PIMs are involved in the pathogenesis of migraine or can differentiate it from non-migraine headaches (nMHs). Using 10 years of data from the Smart Clinical Data Warehouse, patients who visited the neurology outpatient department (OPD) within 30 days after visiting the emergency room (ER) for headaches were divided into migraine and nMH groups, the PIMs were compared including the neutrophil-to-lymphocyte (NLR), monocyte-to-lymphocyte (MLR), platelet-to-lymphocyte (PLR) ratios, and neutrophil-to-monocyte ratio (NMR). Of the 32,761 patients who visited the ER for headaches, 4005 patients visited the neurology OPD within 30 days. There were significant increases in the NLR, MLR, and NMR, but a lower PLR in the migraine and nMH groups than the controls. The NMR was significantly higher in the migraine than the nMH group. A receiver operating characteristic curve analysis showed that the ability of the NLR and NMR to differentiate between migraine and nMHs was poor, whereas it was fair between the migraine groups and controls. The elevated PIMs, particularly the NLR and NMR, during headache attacks in migraineurs suggest that inflammation plays a role in migraine and PIMs may be useful for supporting a migraine diagnosis.

## 1. Introduction

Headache is among the most common neurological symptoms seen in the emergency room (ER). The majority of cases presenting with acute headache attacks in the ER have a diagnosis of primary headache disorders, including migraine and tension-type headache (TTH) [1,2]. With regard to the specific etiologies of primary headaches, migraine is the main condition, representing 93.7% of primary headaches and 55% of the whole headache group seen in the ER [3]. Although not related to mortality, migraine has a high likelihood of generating disability during attacks, leading patients to visit the ER for symptom control. Migraine is a primary headache that affects more than 1 billion people worldwide [4], with a prevalence of approximately 15% in the general population [5]. Additionally, an apparent increase in the prevalence of migraine, but not of other headache types, was found by the Global Burden of Disease Study [6,7]. Although migraine has long been known as a neurological disorder, its underlying pathophysiology is still unclear. Accumulating evidence suggests that it may involve a state of “sterile inflammation” in the intracranial meninges, also called neurogenic inflammation, characterized by the release of neuropeptides, such as neurokinin A, substance P, and calcitonin gene-related peptide from trigeminal innervation, resulting in activation and sensitization of trigeminal meningeal afferent nociceptors [8]. Inflammatory processes are associated with the etiology and clinical progression of neurological disorders, including migraine, psychiatric conditions, epilepsy, cerebrovascular diseases, Alzheimer’s disease, and Parkinson’s disease. Both central and systemic inflammatory actions have been linked to the development of brain diseases, suggesting that complex neuroimmune interactions could contribute to pathological changes in the brain [9]. In addition, epidemiological studies have reported that migraine is comorbid with several chronic inflammatory diseases, including multiple sclerosis, chronic inflammatory rheumatic diseases, and inflammatory bowel diseases. Systemic mediators, such as cytokines and the gut microbiome, can exacerbate or add significant risks to migraine [10]. Based on the concept of neurogenic inflammation, several studies investigated the relations between headaches and various inflammatory markers. Proinflammatory cytokines, such as interleukin (IL)-1β, IL-6, and tumor necrosis factor-alpha (TNF-α), as well as anti-inflammatory cytokines, such as IL-10, are elevated in patients with migraine and are suggested to play a role in migraine [11,12,13,14]. Peripheral blood cells can reflect the inflammatory status of a patient. Particularly, peripheral inflammatory markers (PIMs) based on the differential white blood cell (WBC) count are better predictors of mortality and clinical outcomes in various medical conditions than traditional infection markers, including C-reactive protein (CRP) or total leukocyte count [15,16]. Many peripheral inflammatory biomarkers, including the neutrophil-to-lymphocyte ratio (NLR), monocyte-to-lymphocyte ratio (MLR), platelet-to-lymphocyte ratio (PLR), and neutrophil-to-monocyte ratio (NMR), have been investigated in various headache disorders. To distinguish patients with subarachnoid hemorrhage (SAH) from those with primary headache, one study showed that the NLR was significantly higher in patients with SAH than in those with migraine [17]. Another study reported that neutrophils had more accurate and reliable diagnostic values than the NLR and PLR for distinguishing between SAH and nontraumatic acute headache [18]. In studies that have compared levels of inflammatory markers in migraine patients, serum NLR, MLR, and the CRP-to-albumin ratio were higher in migraine subtypes during an attack, i.e., migraine with aura, and in patients with a family history of migraine [19]. In addition, the NMR was significantly higher in migraine with aura and the PLR and NMR were significantly higher in migraine without aura than in controls during the interictal period [20]. These results suggest that a systemic inflammatory state may be related to the pathogenesis of migraine. In addition, systemic inflammation parameters, including the NLR, PLR, and CRP level, are significantly elevated in chronic TTH patients [21]. However, in a prospective study that investigated the roles of inflammatory markers in migraine and TTH patients during acute headache attack, the NLR did not show a significant difference between groups [22].

A number of attempts have been performed to elucidate the potential relationships of these PIMs with headaches, including migraine. However, it is not yet clear whether these inflammatory markers are related to the pathogenesis of migraine or whether they can be used to differentiate between migraine and non-migraine headaches (nMHs). The present study investigated the utility of PIMs for differentiation between migraine and nMH in patients with an acute headache attack and their potential relationships with migraine.

## 2. Materials and Methods

### 2.1. Study Population

We retrospectively analyzed data from the Smart Clinical Data Warehouse (CDW) of Hallym University Medical Center (HUMC) from January 2012 to February 2022. The Smart CDW, which is based on the QlikView Elite Solution Provider (Qlik, Lund, Sweden), is used at the five hospitals of the HUMC and provides analysis of electronic medical record textual data, as well as integrated analysis of “fixed” data. The enrolled subjects were patients aged ≥20 years with primary headache diagnosis codes at the time of visiting the ER during the approximately 10-year study period (1 January 2012–31 January 2022). We collected clinical data of the patients who visited the neurology outpatient department (OPD) within 30 days after visiting the ER. Patients < 20 years old, >80 years old, those who did not undergo laboratory examination at the time of visiting the ER, and those who did not visit the neurology OPD within 30 days were excluded. Patients with headache were assessed during the attack by emergency medicine specialists or neurologists in the ER and in attack-free periods in the neurology OPD by neurologists. Patients who visited the neurology OPD within 30 days after the ER visit were divided into migraine and nMH groups according to their diagnostic codes. According to the International Classification of Diseases, tenth revision (ICD-10) codes in the database, patients with diagnosis code G43 were classified as belonging to the migraine group and diagnosis codes G44 and R51 were classified as belonging to the nMH group. The control group included patients 20–80 years old who had undergone general health checkups at a health promotion center. Patients < 20 years old, >80 years old, those who did not undergo blood laboratory tests at health checkups, with a history of headache assessed using a basic questionnaire completed prior to the health examination, and patients with a history of headache disorders who visited our medical center were excluded. This study was approved by the Clinical Research Ethics Committee of Chuncheon Sacred Heart Hospital, Hallym University (IRB No. 2022-09-010). As only de-identified data were used in this study, the requirement for informed consent was waived.

### 2.2. Peripheral Inflammatory Markers

Among the enrolled groups, the medical records of subjects with routinely available blood inflammatory markers at the time of the ER visit, including the NLR, MLR, PLR, and NMR, were collected. The NLR was calculated by dividing the number of neutrophils by the number of lymphocytes; the MLR was calculated by dividing the number of monocytes by the number of lymphocytes; the PLR was calculated by dividing the number of platelets by the number of lymphocytes; and the NMR was calculated by dividing the number of neutrophils by the number of monocytes. We compared the data with the PIMs of control subjects without headache determined during health checkups. In addition, comorbidities were defined according to the ICD-10 codes in the database and included diabetes mellitus, hypertension, dyslipidemia, angina, atrial fibrillation, heart disease, cerebrovascular diseases, chronic pulmonary disease, renal failure, chronic hepatitis, anxiety disorder, depression, sleep disorders, and menopause.

### 2.3. Statistical Analysis

Continuous data are presented as the mean and standard deviation (SD) and categorical data are presented as the frequency and percentage. The t test was used for comparison of the continuous data of patients with migraine or nMH. Categorical data were analyzed using the chi-square test. As patients could not be randomized based on the presence of each headache diagnosis, confounding and selection biases were accounted for using propensity scores. Propensity score matching (PSM) was performed between normal controls and the patients with migraine or nMH group and the matching ratio was 1:2. We also performed PSM between the migraine group and the nMH or TTH group and the matching ratio was 1:1. Python (ver. 3.7; Anaconda, Austin, TX, USA) and pymatch (ver. 0.3.4) were used for PSM. We used logistic regression as the estimation algorithm and the nearest neighbor algorithm as the matching algorithm without replacement. The propensity scores ranged from 0.01 to 0.43 using calipers with a width equal to 0.2 of the standard deviation of the logit. After matching, the absolute standardized mean differences (ASD) for the covariates were < 0.05, indicating adequate balance. Receiver operating characteristic (ROC) curve analysis was performed to determine the optimal cutoffs. The Youden index was used to calculate the cutoff values of the PIMs. All *p*-values were two-sided and *p* < 0.05 was taken to indicate statistical significance. SPSS software (ver. 26.0; IBM, Armonk, NY, USA) was used for the statistical analyses.

## 3. Results

### 3.1. Subject Characteristics

The study population included 32,761 patients who had visited the ER for headache between January 2012 and January 2022. In all, 369,670 subjects who underwent health checkups selected from the same database were included as controls. The enrollment process is presented in Figure 1. After excluding 28,756 patients in the headache group and 281,084 subjects in the control group according to the exclusion criteria outlined above, 4005 headache patients who had visited the neurology OPD within 30 days after the ER visit and 88,586 controls without headache were included in the study. The following patients were excluded: 3707 patients < 20 years old, 552 patients > 80 years old, 6155 patients without laboratory tests in the ER, and 18,342 patients who did not visit the neurology OPD within 30 days after the ER visit. Among the headache patients who visited the neurology OPD within 30 days after the ER visit, 1453 patients with migraine and 2552 patients with nMH were identified. After PSM, the identified numbers in each group were as follows: migraine vs. control, 1453 vs. 2906, respectively; nMH vs. control, 2549 vs. 5093, respectively; and migraine vs. nMH, 1410 vs. 1410, respectively (Figure 1).

All absolute standardized differences between the migraine and control group or between the nMH and control group were <0.1 after PSM (Table 1 and Table 2). There were no significant differences in any clinical variables after PSM between the migraine and control group or between the nMH and control groups. In addition, there were no significant differences in any variables after PSM between the migraine and nMH groups (Table 3).

### 3.2. Comparison of Peripheral Inflammatory Markers in Migraine and nMH

Comparison of the PIMs between the migraine and control groups after PSM showed that the NLR (3.06 ± 2.39 vs. 2.13 ± 2.03, respectively, *p* < 0.001), MLR (0.22 ± 0.11 vs. 0.20 ± 0.10, *p* < 0.001), and NMR (14.47 ± 9.41 vs. 10.89 ± 5.34, *p* < 0.001) were higher, while PLR (144.73 ± 66.41 vs. 150.48 ± 61.59, *p* = 0.006) was lower in the migraine group than the control group (Table 4). In addition, in comparisons of PIMs between the nMH and control groups, the NLR (2.61 ± 2.38 vs. 2.01 ± 1.66, *p* < 0.001), MLR (0.23 ± 0.14 vs. 0.20 ± 0.13, *p* < 0.001), and NMR (11.76 ± 7.81 vs. 10.35 ± 4.42, *p* < 0.001) were higher, while the PLR (135.00 ± 65.25 vs. 143.88 ± 64.21, *p* < 0.001) was lower in the nMH group than the control group (Table 5). The NMR was significantly higher in the migraine group than the nMH group (14.01 ± 9.35 vs. 12.13 ± 7.26, *p* < 0.001). The NLR was higher in the migraine group than the nMH group, but the difference did not reach statistical significance (2.99 ± 2.51 vs. 2.70 ± 2.53, *p* = 0.051) (Table 6). In addition, in subgroup analysis between the migraine and TTH groups, the NLR (2.98 ± 2.39 vs. 2.44 ± 2.19, *p* = 0.001) and NMR (13.84 ± 9.34 vs. 11.47 ± 6.45, *p* < 0.001) were significantly higher in the migraine group than the TTH group (Table 7).

### 3.3. Discriminative Ability of Peripheral Inflammatory Markers in Migraine and nMH

Table 8 presents the areas under the curve (AUCs) and optimal cutoff values of PIMs, including the NLR, MLR, PLR, and NMR, for differentiating between the groups (Table 8). The results of the ROC curve analysis showed that the differentiating ability of the NLR and NMR between migraine and control was fair (AUC [95% CI]: NLR 0.65 [0.62–0.67], NMR 0.61 [0.59–0.64], *p* < 0.001). However, the differentiating ability of the NLR and NMR between migraine and nMH was poor (AUC [95% CI]: NLR 0.55 [0.53–0.57], NMR 0.56 [0.54–0.58], *p* < 0.001) (Figure 2).

## 4. Discussion

We retrospectively collected the clinical data of patients who visited the ER with an acute headache attack and analyzed the diagnoses of patients who visited the neurology OPD within 30 days after visiting the ER. Comparison of the PIMs at the ER showed that the NLR, MLR, and NMR were higher in the migraine or nMH group during the acute headache attack than the control group. In comparisons between the primary headache groups, the NLR and NMR were higher in the migraine group than the nMH group. In addition, ROC curve analysis showed that the NLR and NMR had poor ability to differentiate between migraine and nMH, but fair ability to differentiate between migraine and control groups. These findings suggest that a systemic inflammatory state may play a role in migraine and that, although nonspecific, these inflammatory markers may play a supportive role in the diagnosis of migraine.

Many lines of evidence suggest that inflammation is a major contributor to the pathogenesis of various neurological disorders. Particularly, the neurodegenerative processes in Alzheimer’s disease, Parkinson’s disease, stroke, vascular dementia, and multiple sclerosis are fueled by neuroinflammation, which is in turn accompanied by parallel systemic immune dysregulation. This crosstalk between the periphery and brain suggests that the bidirectional flux of inflammatory mediators represents a proactive phenomenon driving these neuropathological processes [23,24,25]. In migraine, there is accumulating evidence that the release of inflammatory agents affects the activation and sensitization of peripheral nociceptors. These inflammatory agents induce the activation of trigeminal nerves and the release of vasoactive neuropeptides, consequently contributing to inflammation [26,27,28]. Proinflammatory cytokines that are responsible for upregulating inflammatory reactions and anti-inflammatory cytokines that control the proinflammatory cytokine response play roles in the regulation of pain [29]. Many studies have attempted to explain the potential relationships of these pro- and anti-inflammatory cytokines with migraine based on their roles in the regulation of pain. The levels of proinflammatory cytokines, such as IL-1β, IL-6, and TNF-α, are elevated in migraineurs compared to controls [12,13,14,30,31,32]. On the other hand, migraineurs have elevated levels of the anti-inflammatory cytokine, IL-10 [11,12]. There is also accumulating evidence for the presence of other abnormal inflammatory markers in the systemic circulation of migraineurs [30,32,33,34,35]. A systemic pro-inflammatory status is the most robust experimental finding on migraines. However, it is still necessary to understand the extent to which the proinflammatory status of migraineurs reduces the threshold for trigeminovascular system activation, which is believed to promote neuroinflammatory events during migraine attacks [36]. Neuroinflammation has an important role in the pathophysiology of migraine, which is a complex neuro-glio-vascular disorder. Recently, the contribution of the activation of the inflammasome, a key component of the innate immune system responsible for multiprotein complex signaling, to the pain signaling system has received attention. Its activation causes the production of inflammatory cytokines that can stimulate trigeminal neurons and are implicated in the generation of migraine pain [37]. Compared to migraine, there have been relatively few studies regarding the relationships between TTH and peripheral inflammation. Increased levels of inflammatory markers, such as IL-8, have been reported in patients with TTH and it was suggested that proinflammatory mechanisms may be involved in the pathophysiology of TTH [38]. However, the use of these inflammatory markers in clinical practice is limited due to both cost and the cumbersome diagnostic procedures required for their determination. Differential WBC counts can be obtained conveniently in clinical practice at no additional cost. Therefore, we used complete blood count (CBC)-derived parameters, including the NLR, MLR, PLR, and NMR measured during ER visits, as PIMs.

In the present study, the NLR, MLR, and NMR were significantly higher during headache attacks in migraine and nMH than in the control group. One study showed that the NLR and MLR were higher during migraine attacks than in controls and that the NLR was higher in migraines with auras than in those without auras [19]. Another study reported a significantly higher NMR in migraineurs with or without auras followed up in the neurology OPD compared to controls, supporting the presence of a continuous inflammatory process even in the interictal periods [20]. Our results are consistent with these previous studies. By contrast, a prospective controlled observational study showed that, although high in headache patients, there were no significant differences in the NLR among migraine, TTH, and controls [22]. Our nMH group included patients with diagnostic codes for TTH or non-specific headache; all types of migraine were excluded. Thus, the nMH group mainly included patients with TTH. In a study of TTH, the NLR in patients with chronic TTH was significantly higher than in those with frequent episodic TTH and the control group [21]. A previous study showed that knowledge of the different profiles of inflammatory markers can shed light on the pathophysiological differences between migraine and TTH. As different pathophysiological mechanisms have been proposed for migraine and TTH, the molecular mechanisms might also be distinct [39]. Increased tenderness to palpation of the pericranial myofascial tissues, indicative of peripheral pain mechanisms, is the most obvious abnormality in patients with TTH. Central neuroplastic changes may affect the regulation of peripheral mechanisms and lead to increased pericranial muscle activity or neurotransmitter release in myofascial tissues [40]. Thus, the increase in PIMs in our nMH group, including TTH, is thought to be due to a mechanism different from that in the migraine group. In particular, elevated PIMs in TTH may be associated with a common peripheral inflammatory reaction originating from the myofascial tissue. However, the PLR was significantly lower in patients with migraine and nMH during attacks than the control group in the present study. Transient elevation of the serum WBC count and decreases in platelet count are normal physiological reactions to inflammation [41]. The potential relationship between platelet biology and headache, particularly migraine, has been studied for many years. Despite clinical evidence that patients with essential thrombocythemia have a high frequency of headache, epidemiological studies that have investigated the platelet count in patients with migraine failed to find significant associations [36]. In addition, a possible link between platelet biology and migraine is represented by inflammation. After the formation of platelet–leukocyte aggregates, the increased release of several proinflammatory cytokines may occur, and these mediators may contribute to further increases in neurogenic inflammation in the brain [42,43]. In contrast to the results of the present study, previous studies have reported that the PLR was significantly higher in patients with migraine during attacks and migraine without aura compared to controls [19,20]. In addition, the PLR was higher in the patients with TTH included in the nMH group than the controls in the present study [19]. Further research is needed to understand the association between the PLR and headache, particularly migraine.

In addition, comparison between primary headache subtypes showed that the NLR and NMR were significantly higher in migraine patients than in those with nMH or TTH. However, the ability of the NLR and NMR to differentiate between migraine and nMH was poor, but was fair between migraine and control, as shown by ROC curve analysis. The NLR is an important indicator of peripheral inflammation and oxidative stress associated with chronic neurological diseases [44,45,46,47,48]. Neuroinflammation is associated with the pathogenesis of many neurological diseases, including neurodegenerative diseases. Chronic inflammation plays an important role in the emergence and progression of chronic neurodegenerative diseases of the brain, such as Parkinson’s disease and Alzheimer’s disease [48]. Acute inflammation arises as a result of the migration of neutrophils and macrophages to the site of inflammation mediated by the release of cytokines and chemokines, whereas chronic inflammation is caused by macrophages, lymphocytes, and plasma cell infiltration. An increased neutrophil count is often associated with the occurrence, progression, and severity of inflammation, while a decreased lymphocyte count, as part of the immune regulatory barrier, is associated with the body’s stress response [49,50]. Therefore, the NLR combines information derived from two different pathways, i.e., the neutrophils responsible for ongoing inflammation and the lymphocytes representing immune regulatory pathways [51]. Many previous studies have shown that the NLR is associated with systemic inflammation and chronic disease and demonstrated elevated NLR in neurodegenerative diseases as well as chronic diseases; the NLR has a significant association with prevalent chronic conditions, including hypertension and diabetes [52]. The NLR is also associated with mortality and prognosis in chronic progressive diseases, such as chronic renal disease and peripheral vascular disease due to atherosclerosis. Previous studies showed that a high NLR is a strong predictor of arteriovenous fistula maturation failure, early thrombosis, and mortality in end-stage renal disease [53,54] and also predicts a poor outcome in patients with surgically revascularized peripheral vascular disease [55,56]. Therefore, the NLR is an appropriate marker for use in studies of migraine, a chronic and progressive disease.

More recently, the NMR has been used as a prognostic indicator of malignancy and systemic inflammation [57,58], but there have been few reports regarding the NMR in headache. In the only such study to date in migraine, the NMR was higher in migraine patients with or without auras than in controls, although the differences were not statistically significant [20]. Only a few studies have investigated the correlations between CBC-derived inflammatory markers, particularly the NMR, and classifications of primary headache or subtypes of migraine. Further comprehensive studies are needed to evaluate the roles of these PIMs in headache, particularly migraine.

This study had some limitations. First, unmeasured confounders may not have been accounted for due to the retrospective study design. Second, the data were from subjects who visited one university medical center with five hospitals, so it is difficult to generalize the results to the general population and there is a possibility of selection bias. Third, detailed clinical information on headache characteristics was not collected due to the retrospective nature of the study. In addition, we collected the results of only one blood test performed in the ER. Further studies of the correlations between PIMs and clinical characteristics of patients with headache during ictal and interictal periods are needed.

## 5. Conclusions

In conclusion, migraineurs showed elevated PIMs, particularly the NLR and NMR, during acute headache attacks and the ability of the NLR and NMR to differentiate between migraine and control was fair as shown by the ROC curve analysis. These findings suggest that systemic inflammation may play a role in the etiology of migraine and these inflammatory markers may be useful for supporting a diagnosis of migraine. The NLR, a parameter associated with systemic inflammation and chronic disease, is considered a suitable marker for studies in migraine patients. However, further studies regarding the NMR in headache patients are needed. Further prospective, population-based studies are needed to determine the relationships between headache, particularly migraine, and PIMs.

## Figures and Tables

**Figure 1 jcm-11-06538-f001:**
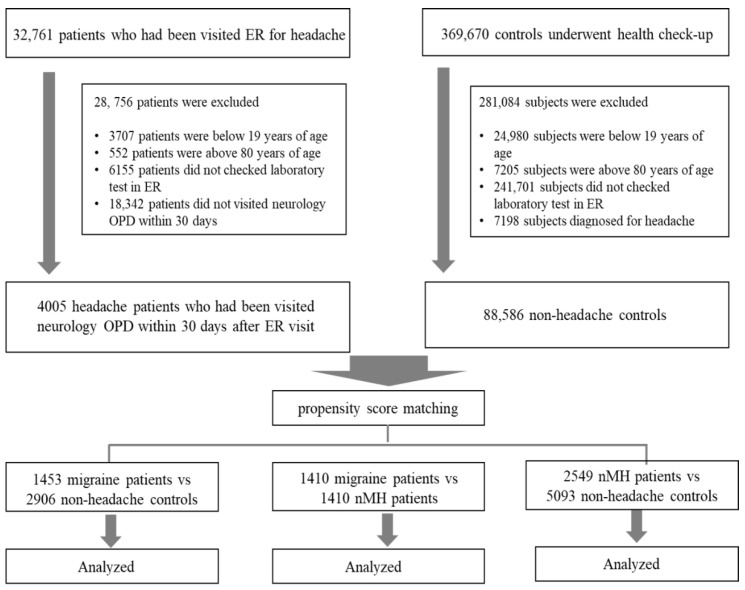
Flow chart of the enrollment process. ER, emergency room; nMH, non-migraine headache; OPD, outpatient department.

**Figure 2 jcm-11-06538-f002:**
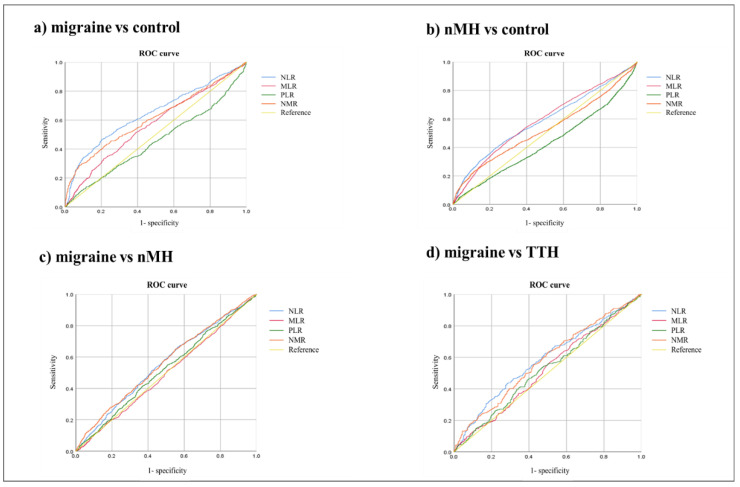
ROC curves of peripheral inflammatory markers for differentiating between groups. ROC, receiver operating characteristic; NLR. neutrophil-to-lymphocyte ratio; MLR, monocyte-to-lymphocyte ratio; PLR, platelet-to-lymphocyte ratio; NMR, lymphocyte-to-monocyte ratio; nMH, non-migraine headache; TTH, tension-type headache.

**Table 1 jcm-11-06538-t001:** Clinical characteristics of the migraine and control groups before and after PSM.

	Before PSM			After PSM		
	Migraine	Control	ASD	Migraine	Control	ASD
	(*n* = 1453)	(*n* = 88,586)		(*n* = 1453)	(*n* = 2906)	
Female (*n*, %)	1065 (73.3%)	39,380 (44.5%)	0.652	1065 (73.3%)	2104 (72.4%)	0.020
Age (mean, SD)	41.8 ± 12.8	45.9 ± 12.9	0.321	41.8 ± 12.8	42.3 ± 13.1	0.042
DM (*n*, %)	40 (2.8%)	4817 (5.4%)	0.164	40 (2.8%)	80 (2.8%)	<0.001
HTN (*n*, %)	136 (9.4%)	7196 (8.1%)	0.042	136 (9.4%)	291 (10.0%)	0.022
Dyslipidemia (*n*, %)	105 (7.2%)	10,019 (11.2%)	0.158	105 (7.2%)	208 (7.2%)	0.003
Angina (*n*, %)	71 (4.9%)	2365 (2.7%)	0.103	71 (4.9%)	156 (5.4%)	0.022
AF (*n*, %)	12 (0.8%)	632 (0.7%)	0.012	12 (0.8%)	28 (1.0%)	0.015
Heart disease (*n*, %)	62 (4.3%)	2012 (2.3%)	0.099	62 (4.3%)	148 (5.1%)	0.041
CVD (*n*, %)	160 (11.0%)	3135 (3.5%)	0.239	160 (11.0%)	334 (11.5%)	0.015
Pulmonary dis (*n*, %)	79 (5.4%)	3853 (4.3%)	0.048	79 (5.4%)	143 (4.9%)	0.023
Renal failure (*n*, %)	13 (0.9%)	817 (0.9%)	0.003	13 (0.9%)	32 (1.1%)	0.022
Chronic hepatitis (*n*, %)	23 (1.6%)	4156 (4.7%)	0.249	23 (1.6%)	39 (1.3%)	0.019
Anxiety (*n*, %)	47 (3.2%)	963 (1.1%)	0.121	47 (3.2%)	97 (3.3%)	0.006
Depression (*n*, %)	101 (7.0%)	1989 (2.2%)	0.185	101 (7.0%)	236 (8.1%)	0.046
Sleep disorder (*n*, %)	68 (4.7%)	1793 (2.0%)	0.126	68 (4.7%)	134 (4.6%)	0.003
Menopause (*n*, %)	32 (2.2%)	1963 (2.2%)	0.001	32 (2.2%)	71 (2.4%)	0.016

AF, atrial fibrillation; ASD, absolute standardized difference; CVD, cerebrovascular disease; DM, diabetes mellitus; HTN, hypertension; PSM, propensity score matching; SD, standard deviation.

**Table 2 jcm-11-06538-t002:** Clinical characteristics of the non-migraine headache and control groups before and after PSM.

	Before PSM			After PSM		
	nMH	Control	ASD	nMH	Control	ASD
	(*n* = 2552)	(*n* = 88,586)		(*n* = 2549)	(*n* = 5093)	
Female (*n*, %)	1513 (59.3%)	39,380 (44.5%)	0.302	1510 (59.2%)	2932 (57.6%)	0.033
Age (mean, SD)	47.9 ± 13.6	45.9 ± 12.9	0.150	47.9 ± 13.6	48.5 ± 13.8	0.043
DM (*n*, %)	135 (5.3%)	4817 (5.4%)	0.007	135 (5.3%)	305 (6.0%)	0.031
HTN (*n*, %)	362 (14.2%)	7196 (8.1%)	0.174	361 (14.2%)	750 (14.7%)	0.016
Dyslipidemia (*n*, %)	201 (7.9%)	10,019 (11.3%)	0.128	200 (7.8%)	426 (8.4%)	0.019
Angina (*n*, %)	218 (8.5%)	2365 (2.7%)	0.210	215 (8.4%)	432 (8.5%)	0.005
AF (*n*, %)	21 (0.8%)	632 (0.7%)	0.012	21 (0.8%)	27 (0.5%)	0.033
Heart disease (*n*, %)	125 (4.9%)	2012 (2.3%)	0.122	125 (4.9%)	250 (4.9%)	<0.001
CVD (*n*, %)	353 (13.8%)	3135 (3.5%)	0.298	350 (13.7%)	742 (14.6%)	0.027
Pulmonary dis (*n*, %)	141 (5.5%)	3853 (4.3%)	0.052	140 (5.5%)	276 (5.4%)	0.003
Renal failure (*n*, %)	32 (1.3%)	817 (0.9%)	0.030	32 (1.3%)	89 (1.7%)	0.044
Chronic hepatitis (*n*, %)	77 (3.0%)	4156 (4.7%)	0.098	77 (3.0%)	138 (2.7%)	0.017
Anxiety (*n*, %)	112 (4.4%)	963 (1.1%)	0.161	109 (4.3%)	192 (3.8%)	0.024
Depression (*n*, %)	162 (6.3%)	1989 (2.2%)	0.168	159 (6.2%)	318 (6.2%)	0.002
Sleep disorder (*n*, %)	110 (4.3%)	1793 (2.0%)	0.113	109 (4.3%)	204 (4.0%)	0.014
Menopause (*n*, %)	64 (2.5%)	1963 (2.2%)	0.019	64 (2.5%)	128 (2.5%)	0.004

AF, atrial fibrillation; ASD, absolute standardized difference; CVD, cerebrovascular disease; DM, diabetes mellitus; HTN, hypertension; nMH, non-migraine headache; PSM, propensity score matching; SD, standard deviation.

**Table 3 jcm-11-06538-t003:** Clinical characteristics of the migraine and non-migraine headache groups before and after PSM.

	Before PSM			After PSM		
	Migraine	nMH	ASD	Migraine	nMH	ASD
	(*n* = 1453)	(*n* = 2552)		(*n* = 1410)	(*n* = 1410)	
Female (*n*, %)	1065 (73.3%)	1513 (59.3%)	0.317	1022 (72.5%)	1005 (71.3%)	0.027
Age (mean, SD)	41.8 ± 12.8	47.9 ± 13.6	0.479	42.2 ± 12.8	42.7 ± 12.5	0.040
DM (*n*, %)	40 (2.8%)	135 (5.3%)	0.155	40 (2.8%)	37 (2.6%)	0.013
HTN (*n*, %)	136 (9.4%)	362 (14.2%)	0.166	136 (9.6%)	134 (9.5%)	0.005
Dyslipidemia (*n*, %)	105 (7.2%)	201 (7.9%)	0.025	104 (7.4%)	98 (7.0%)	0.016
Angina (*n*, %)	71 (4.9%)	218 (8.5%)	0.170	71 (5.0%)	67 (4.8%)	0.013
AF (*n*, %)	12 (0.8%)	21 (0.8%)	<0.001	12 (0.9%)	14 (1.0%)	0.016
Heart disease (*n*, %)	62 (4.3%)	125 (4.9%)	0.031	61 (4.3%)	58 (4.1%)	0.011
CVD (*n*, %)	160 (11.0%)	353 (13.8%)	0.090	160 (11.3%)	157 (11.1%)	0.007
Pulmonary dis (*n*, %)	79 (5.4%)	141 (5.5%)	0.004	79 (5.6%)	65 (4.6%)	0.044
Renal failure (*n*, %)	13 (0.9%)	32 (1.3%)	0.038	13 (0.9%)	12 (0.9%)	0.008
Chronic hepatitis (*n*, %)	23 (1.6%)	77 (3.0%)	0.115	23 (1.6%)	20 (1.4%)	0.017
Anxiety (*n*, %)	47 (3.2%)	112 (4.4%)	0.065	46 (3.3%)	48 (3.4%)	0.008
Depression (*n*, %)	101 (7.0%)	162 (6.3%)	0.024	100 (7.1%)	94 (6.7%)	0.017
Sleep disorder (*n*, %)	68 (4.7%)	110 (4.3%)	0.018	65 (4.6%)	53 (3.8%)	0.040
Menopause (*n*, %)	32 (2.2%)	64 (2.5%)	0.021	32 (2.3%)	28 (2.0%)	0.019

AF, atrial fibrillation CVD, cerebrovascular disease; ASD, absolute standardized difference; DM, diabetes mellitus; HTN, hypertension; nMH, non-migraine headache; PSM, propensity score matching; SD, standard deviation.

**Table 4 jcm-11-06538-t004:** Peripheral inflammatory markers of the migraine and control groups after PSM.

	Migraine	Control	*p*-Value
	(*n* = 1453)	(*n* = 2906)	
Neutrophil count	5.04 ± 2.29	3.50 ± 1.70	<0.001
Monocyte count	0.40 ± 0.16	0.34 ± 0.13	<0.001
Lymphocyte count	2.04 ± 0.88	1.87 ± 0.60	<0.001
Platelet count	255.40 ± 63.28	256.97 ± 59.25	0.106
NLR	3.06 ± 2.39	2.13 ± 2.03	<0.001
MLR	0.22 ± 0.11	0.20 ± 0.10	<0.001
PLR	144.73 ± 66.41	150.48 ± 61.59	0.006
NMR	14.47 ± 9.41	10.89 ± 5.34	<0.001

MLR, monocyte-to-lymphocyte ratio; NLR, neutrophil-to-lymphocyte ratio; NMR, neutrophil-to-monocyte ratio; PLR, platelet-to-lymphocyte ratio; PSM, propensity score matching.

**Table 5 jcm-11-06538-t005:** Peripheral inflammatory markers of the non-migraine headache and control groups after PSM.

	nMH	Control	*p*-Value
	(*n* = 2549)	(*n* = 5093)	
Neutrophil count	4.48 ± 2.23	3.47 ± 1.66	<0.001
Monocyte count	0.43 ± 0.19	0.36 ± 0.76	0.100
Lymphocyte count	2.10 ± 0.85	1.93 ± 0.63	<0.001
Platelet count	246.25 ± 61.41	254.20 ± 63.12	0.609
NLR	2.61 ± 2.38	2.01 ± 1.66	<0.001
MLR	0.23 ± 0.14	0.20 ± 0.13	<0.001
PLR	135.00 ± 65.25	143.88 ± 64.21	<0.001
NMR	11.76 ± 7.81	10.35 ± 4.42	<0.001

MLR, monocyte-to-lymphocyte ratio; NLR, neutrophil-to-lymphocyte ratio; nMH, non-migraine headache; NMR, neutrophil-to-monocyte ratio; PLR, platelet-to-lymphocyte ratio; PSM, propensity score matching.

**Table 6 jcm-11-06538-t006:** Peripheral inflammatory markers of the migraine and non-migraine headache groups after PSM.

	Migraine	nMH	*p*-Value
	(*n* = 1410)	(*n* = 1410)	
Neutrophil count	4.92 ± 2.31	4.58 ± 2.32	0.214
Monocyte count	0.41 ± 0.17	0.42 ± 0.18	0.223
Lymphocyte count	2.05 ± 0.87	2.10 ± 0.84	0.894
Platelet count	252.36 ± 62.47	253.36 ± 62.28	0.794
NLR	2.99 ± 2.51	2.70 ± 2.53	0.051
MLR	0.22 ± 0.11	0.23 ± 0.13	0.074
PLR	142.93 ± 69.35	139.40 ± 68.99	0.566
NMR	14.01 ± 9.35	12.13 ± 7.26	<0.001

MLR, monocyte-to-lymphocyte ratio; NLR, neutrophil-to-lymphocyte ratio; nMH, non-migraine headache; NMR, neutrophil-to-monocyte ratio.; PLR, platelet-to-lymphocyte ratio; PSM, propensity score matching.

**Table 7 jcm-11-06538-t007:** Peripheral inflammatory markers of the migraine and TTH groups after PSM.

	Migraine	TTH	*p*-Value
	(*n* = 458)	(*n* = 458)	
Neutrophil count	4.99 ± 2.34	4.35 ± 2.21	0.041
Monocyte count	0.41 ± 0.17	0.42 ± 0.18	0.483
Lymphocyte count	2.07 ± 0.85	2.12 ± 0.80	0.168
Platelet count	252.72 ± 59.24	255.42 ± 66.28	0.168
NLR	2.98 ± 2.39	2.44 ± 2.19	0.001
MLR	0.22 ± 0.11	0.21 ± 0.10	0.467
PLR	140.42 ± 62.48	135.81 ± 61.56	0.505
NMR	13.84 ± 9.34	11.47 ± 6.45	<0.001

MLR, monocyte-to-lymphocyte ratio; NLR, neutrophil-to-lymphocyte ratio; NMR, neutrophil-to-monocyte ratio; PLR, platelet-to-lymphocyte ratio; PSM, propensity score matching; TTH, tension-type headache.

**Table 8 jcm-11-06538-t008:** AUC and optimal cutoff values of inflammatory markers for differentiating between groups.

	AUC (95% CI)	Cutoff Value	Sensitivity	Specificity	*p*-Value
Migraine vs. control					
NLR	0.65 (0.62–0.67)	2.45	0.47	0.80	<0.001
MLR	0.57 (0.54–0.60)	0.19	0.51	0.61	<0.001
PLR	0.45 (0.43–0.48)	206.95	0.14	0.88	<0.001
NMR	0.61 (0.59–0.64)	16.49	0.28	0.92	<0.001
nMH vs. control					
NLR	0.59 (0.57–0.60)	2.29	0.39	0.78	<0.001
MLR	0.59 (0.57–0.60)	0.21	0.44	0.71	<0.001
PLR	0.43 (0.42–0.44)	261.55	0.05	0.97	<0.001
NMR	0.53 (0.51–0.54)	14.23	0.24	0.88	<0.001
Migraine vs. nMH					
NLR	0.55 (0.53–0.57)	1.72	0.67	0.44	<0.001
MLR	0.49 (0.47–0.52)	0.10	0.97	0.04	0.579
PLR	0.52 (0.50–0.54)	141.12	0.41	0.63	0.071
NMR	0.56 (0.54–0.58)	9.50	0.65	0.45	<0.001
Migraine vs. TTH					
NLR	0.58 (0.55–0.62)	2.50	0.43	0.72	<0.001
MLR	0.52 (0.48–0.56)	0.17	0.69	0.37	0.293
PLR	0.53 (0.49–0.56)	132.97	0.47	0.60	0.197
NMR	0.58 (0.54–0.62)	9.48	0.64	0.49	<0.001

nMH, non-migraine headache; TTH, tension-type headache; NLR, neutrophil-to-lymphocyte ratio; MLR, monocyte-to-lymphocyte ratio; PLR, platelet-to-lymphocyte ratio; NMR, neutrophil-to-monocyte ratio.

## Data Availability

The original contributions presented in the study are included in the article, further inquiries can be directed to the corresponding author.

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
