# Peer review of "Role of Peripheral Inflammatory Markers in Patients with Acute Headache Attack to Differentiate between Migraine and Non-Migraine Headache"

_jcm, 2022, doi:10.3390/jcm11216538_

Round 1

Reviewer 1 Report

I would like to congratulate the authors on their work. The manuscript is a neat study about the role of peripheral inflammatory markers in patients with acute headache attack to differentiate between migraine and non-migraine headache.

1. Abstract:

Please re-write the Abstract conform to the requirements:” Background: Place the question addressed in a broad context and highlight the purpose of the study; Methods: Describe briefly the main methods or treatments applied. Include any relevant preregistration numbers, and species and strains of any animals used. Results: Summarize the article's main findings; Conclusion: Indicate the main conclusions or interpretations.”

2. Materials and methods:

2.1 I suggest the authors to delete all the codes included in subsection “2.2. Peripheral Inflammatory Markers”, it doesn’t add value to the paper and it’s not actually relevant. Just keep the comorbidities list.

2.2 Please delete “(version 0.3.4; https://github.com/benmiroglio/pymatch)” and just keep the version.

2.3 Please use a dot when giving a number, not a comma, to make sure it’s not confusing.

3. Discussion:

3.1 Line 330: delete “(47, 48)

3.2 I highly recommend the authors to improve the quality of the research by comparing the results regarding the NLR, MLR, and PLR with other articles in the Discussion section regarding other disease, like cardiovascular disease, end stage kidney disease, and other chronic condition. Please add and compare the results with the following articles:

-https://doi.org/10.3390/life12091447

-https://doi.org/10.3390/biomedicines10061272

-https://doi.org/10.3390/jcm11092620

-https://doi.org/10.3390/life12060822

Kind regards

Author Response

Nov 1, 2022 Reviewer 1 Journal of Clinical Medicine Dear Reviewer 1, Please find attached a revised version of our manuscript, “Role of Peripheral Inflammatory Markers in Patients with Acute Headache Attack to Differentiate Between Migraine and Non-Migraine Headache” (jcm-2007998). We thank you for your thoughtful suggestions regarding the original version of our paper; most of the suggested changes have been incorporated into the revision. All revisions are described in detail in the order mentioned in the review, following your comment (in italics). We believe that the revisions have greatly improved the manuscript and hereby submit the revised version for your consideration for publication. Comments to author: I would like to congratulate the authors on their work. The manuscript is a neat study about the role of peripheral inflammatory markers in patients with acute headache attack to differentiate between migraine and non-migraine headache. We thank the reviewer for these comments and suggestions, which have improved our manuscript. 1. Abstract: Please re-write the Abstract conform to the requirements:” Background: Place the question addressed in a broad context and highlight the purpose of the study; Methods: Describe briefly the main methods or treatments applied. Include any relevant preregistration numbers, and species and strains of any animals used. Results: Summarize the article's main findings; Conclusion: Indicate the main conclusions or interpretations.” Following the guidelines of Journal of Clinical Medicine (“Abstract: The abstract should be a total of about 200 words maximum. The abstract should be a single paragraph and should follow the style of structured abstracts, but without headings”), we have revised the Abstract as follows: Although the potential relationship between headache, particularly migraine, and peripheral inflammatory markers (PIMs) has been investigated, it is unclear whether PIMs are involved in the pathogenesis of migraine or can differentiate it from non-migraine headache (nMH). Using 10 years of data from the Smart Clinical Data Warehouse, patients who visited the neurology outpatient department (OPD) within 30 days after visiting the emergency room (ER) for headache were divided into migraine and nMH groups. The PIMs were compared including the neutrophil-to-lymphocyte (NLR), monocyte-to-lymphocyte (MLR), platelet-to-lymphocyte (PLR) ratios, and neutrophil-to-monocyte ratio (NMR). Of the 32,761 patients who visited the ER for headache, 4,005 patients visited the neurology OPD within 30 days. There were significant increases in NLR, MLR, and NMR, but lower PLR in the migraine and nMH groups than the controls. The NMR was significantly higher in the migraine than nMH group. Receiver operating characteristic curve analysis showed that the ability of NLR and NMR to differentiate between migraine and nMH was poor, whereas it was fair between the migraine groups and controls. The elevated PIMs, particularly NLR and NMR, during headache attacks in migraineurs suggests that inflammation plays a role in migraine and PIMs may be useful for supporting a migraine diagnosis. (page 1, lines 15 – page 1, lines 35) 2. Materials and methods: 2.1 I suggest the authors to delete all the codes included in subsection “2.2. Peripheral Inflammatory Markers”, it doesn’t add value to the paper and it’s not actually relevant. Just keep the comorbidities list. As recommended, we have deleted all codes in the Materials and Methods section, as follows: In addition, comorbidities were defined according to the ICD-10 codes in the database, and included diabetes mellitus, hypertension, dyslipidemia, angina, atrial fibrillation, heart disease, cerebrovascular diseases, chronic pulmonary disease, renal failure, chronic hepatitis, anxiety disorder, depression, sleep disorders, and menopause. (page 3, lines 137 – page 3, lines 144) 2.2 Please delete “(version 0.3.4; https://github.com/benmiroglio/pymatch)” and just keep the version. We have deleted this from the Materials and Methods, as follows: Python (ver. 3.7; Anaconda, Austin, TX, USA) and pymatch (ver. 0.3.4) were used for PSM. (page 4, lines 154 – page 4, lines 156) 2.3 Please use a dot when giving a number, not a comma, to make sure it’s not confusing. We have revised this as suggested. 3. Discussion: 3.1 Line 330: delete “(47, 48)” We apologize for the mistake. We have deleted this. 3.2 I highly recommend the authors to improve the quality of the research by comparing the results regarding the NLR, MLR, and PLR with other articles in the Discussion section regarding other disease, like cardiovascular disease, end stage kidney disease, and other chronic condition. Please add and compare the results with the following articles: -https://doi.org/10.3390/life12091447 -https://doi.org/10.3390/biomedicines10061272 -https://doi.org/10.3390/jcm11092620 -https://doi.org/10.3390/life12060822 Thank you. We have added text to the Discussion as follows: NLR is also associated with mortality and prognosis in chronic progressive diseases, such as chronic renal disease and peripheral vascular disease due to atherosclerosis. Previous studies showed that a high NLR is a strong predictor of arteriovenous fistula maturation failure, early thrombosis, and mortality in end-stage renal disease [53,54] and also predicts a poor outcome in patients with surgically revascularized peripheral vascular disease [55,56]. (page 11, lines 367 – page 11, lines 372) We have also added these citations. 53. Muresan AV.; Russu E.; Arbănasi EM.; Kaller R.; Hosu I.; Arbănasi EM.;Septimiu Toader Voidăzan The Predictive Value of NLR, MLR, and PLR in the Outcome of End-Stage Kidney Disease Patients. Biomedicines 2022, 10, 1272. 54. Kaller, R.; Arbănasi, EM.; Muresan, AV.; Voidăzan, S.; Arbănasi, EM.; Horváth, E.; Suciu, BA.; Hosu, L.; Halmaciu, L.; Brinzaniuc, K.; Russu, E. The Predictive Value of Systemic Inflammatory Markers, the Prognostic Nutritional Index, and Measured Vessels’ Diameters in Arteriovenous Fistula Maturation Failure. Life 2022, 12(9), 1447. 55. Russu, E.; Muresan, AV.; Arbănasi, EM.; Kaller, R.; Hosu, L.; Voidăzan, S.; Arbanasi EM.; Cosarcă CM. The Predictive Role of NLR and PLR in Outcome and Patency of Lower Limb Revascularization in Patients with Femoropopliteal Dis-ease. J. Clin. Med. 2022, 11, 2620. 56. Arbănasi, EM.; Muresan AV.; Cosarcă, CM.; Kaller, R.; Bud, TL.; Hosu, L.; Voidăzan, ST.; EArbănasi EM.; Russu, E. Neu-trophil-to-Lymphocyte Ratio and Platelet-to-Lymphocyte Ratio Impact on Predicting Outcomes in Patients with Acute Limb Ischemia. Life 2022, 12, 822. Kind regards We have addressed all of the issues raised by the reviewers. We are grateful for the constructive comments made during the review process. We believe that our paper has been improved by implementing these suggestions. Yours faithfully, Jong-Hee Sohn, M.D. Ph.D. Department of Neurology, Chuncheon Sacred Heart Hospital, Hallym University College of Medicine, 77 Sakju-ro, Chuncheon-si, Gangwon-do, 24253, Republic of Korea Tel: +82-33-252-9970, Fax: +82-33-241-8063 E-mail: deepfoci@hallym.or.kr

Reviewer 2 Report

The authors address an interesting topic in the area of headaches, the study of peripheral markers of inflammation in migraine. This area has already been studied in the past but the most recent hypotheses on the role of the inflamasome in migraine make it still relevant today.

Insert where appropriate PMID: 34112082 

The introduction refers to outdated publications.

Replace ref 1 and 2 with PMID: 32169031 PMID: 31690261 

Reference 5 should be updated with a more recent and complete reference by the same author: PMID: 33267788 

An introductory discussion on the importance of headache epidemiology is missing. Insert and discuss: PMID: 35410119 

The neuroimmunological view of migraine should be mentioned and discussed in the final discussion: PMID: 35020858 

Author Response

Nov 1, 2022 Reviewer 2 Journal of Clinical Medicine Dear Reviewer 2, Please find attached a revised version of our manuscript, “Role of Peripheral Inflammatory Markers in Patients with Acute Headache Attack to Differentiate Between Migraine and Non-Migraine Headache” (jcm-2007998). We thank you for your thoughtful suggestions regarding the original version of our paper; most of the suggested changes have been incorporated into the revision. All revisions are described in detail in the order mentioned in the review, following your comment (in italics). We believe that the revisions have greatly improved the manuscript and hereby submit the revised version for your consideration for publication. Comments to author: The authors address an interesting topic in the area of headaches, the study of peripheral markers of inflammation in migraine. This area has already been studied in the past but the most recent hypotheses on the role of the inflamasome in migraine make it still relevant today. Insert where appropriate PMID: 34112082 We thank the reviewer for these comments and suggestions, which have improved our manuscript. We have added text to the Discussion as follows: Neuroinflammation has an important role in the pathophysiology of migraine, which is a complex neuro-glio-vascular disorder. Recently, the contribution of activation of the inflammasome, a key component of the innate immune system responsible for multiprotein complex signaling, to the pain signaling system has received attention. Its activation causes the production of inflammatory cytokines that can stimulate trigeminal neurons and are implicated in the generation of migraine pain.[37]. (page 10, lines 291 – page 10, lines 296) We have also added the following citation. 37. Kursun O, Yemisci, M.; Maagdenberg, AM.; Karatas H. Migraine and neuroinflammation: the inflammasome perspective. The Journal of Headache and Pain 2021, 22,55. https://doi.org/10.1186/s10194-021-01271-1 The introduction refers to outdated publications. Replace ref 1 and 2 with PMID: 32169031 PMID: 31690261 As recommended, we have replaced references 1 and 2 with the following: 1. Negro, A.; Spuntarelli, V.; Sciattella, P.; Martelletti, P. Rapid referral for headache management from emergency department to headache centre: four years data The Journal of Headache and Pain. 2020, 21.25. https://doi.org/10.1186/s10194-020-01094-6. Peters, K.S. Secondary headache and head pain emergencies. Primary care 2004, 31, 381-393, vii, doi:10.1016/j.pop.2004.02.009. 2. Doretti, A.; Shestaritc, I.; Ungaro, D.; Lee, J.; Lymperopoulos, L.; Kokoti, L.; Guglielmetti, M.; Mitsikostas, DD.; Lampl, C.; and on behalf of the School of Advanced Studies of the European Headache Federation (EHF-SAS). Headaches in the emergency department–a survey of patients’ characteristics, facts and needs. The Journal of Headache and Pain 2019, 20,100. https://doi.org/10.1186/s10194-019-1053-5. Reference 5 should be updated with a more recent and complete reference by the same author: PMID: 33267788 As recommended, we replaced reference 5 with the following: 5. Steiner, TJ.; Stovner, LJ.; Jensen, R.; Uluduz, D.; Katsarava. Z.; on behalf of Lifting The Burden: the Global Campaign against Headache. Migraine remains second among the world’s causes of disability, and first among young women: findings from GBD2019. The Journal of Headache and Pain 2020,21,137. https://doi.org/10.1186/s10194-020-01208-0. An introductory discussion on the importance of headache epidemiology is missing. Insert and discuss: PMID: 35410119 Thank you. We have added text to the Introduction as follows: Also, an apparent increase in the prevalence of migraine, but not of other headache types, was found by the Global Burden of Disease Study [6.7]. (page 2, lines 48 – page 2, lines 50) We have also added these citations. 6. Stovner, LJ.; Hagen, K.; Jensen, R.; Katsarava, Z.; Lipton, R.; Scher, A.; Steiner, T; Zwart JA. (2007) The global burden of headache: a documentation of headache prevalence and disability worldwide. Cephalalgia 27,193–210. 7. Stovner, LJ.; Hagen, K.; Linde, M.; Steiner, TJ. The global prevalence of headache: an update, with analysis of the influ-ences of methodological factors on prevalence estimates. The Journal of Headache and Pain, 2022, 23, 34. https://doi.org/10.1186/s10194-022-01402-2. The neuroimmunological view of migraine should be mentioned and discussed in the final discussion: PMID: 35020858 We have added text to the Discussion as follows: Systemic pro-inflammatory status is the most robust experimental finding in migraine. However, it is still necessary to understand the extent to which the proinflammatory status of migraineurs reduces the threshold for trigeminovascular system activation, which is believed to promote neuroinflammatory events during migraine attacks [36]. (page 10, lines 287 – page 10, lines 291) We also added this citation. 36. Biscetti, L.; Vanna, GD.; Cresta, E.; Bellotti, A.; Corbelli, L.; Cupini, ML.; Calabresi, P.; Sarchielli, P. Immunological findings in patients with migraine and other primary headaches: a narrative review. Clinical and Experimental Immunology, 2022, 207, 11–26. https://doi.org/10.1093/cei/uxab025. We have addressed all of the issues raised by the reviewers. We are grateful for the constructive comments made during the review process. We believe that our paper has been improved by implementing these suggestions. Yours faithfully, Jong-Hee Sohn, M.D. Ph.D. Department of Neurology, Chuncheon Sacred Heart Hospital, Hallym University College of Medicine, 77 Sakju-ro, Chuncheon-si, Gangwon-do, 24253, Republic of Korea Tel: +82-33-252-9970, Fax: +82-33-241-8063 E-mail: deepfoci@hallym.or.kr

Reviewer 3 Report

The purpose of the article was to investigate the role of peripheral inflammatory markers (PIMs) in differentiating migraine from non-migraine headache (nMH) in patients with
acute headache attacks. A retrospective study over a ten-year period. The study showed that PIMs were pathologic compared to controls in patients with migraine as well as in patients with nMH. However, the two patient groups have some similarities.

Absolutely, a well-done study and an interesting one. I have no serious comments to introduction, materials and methods, as well as results. Only, I think it could be valuable that a statistician took an overview for control of PSM.

Discussion: I miss some considerations about PIMs of the nMH group. I assume that most of the nMH patients suffer from tension-type headache (TTH). Probably, TTH suffer from a simple muscular inflammation. In rows 247 – 51: Poor ability of NMR and NLR to differentiate between migraine and nMH, but fair between migraine and control. Thus, migraine patients and nMH patients have similar PIMs findings. However, they do not necessarily have the same cause of the findings. I guess there can be different causes of the PIMs findings. TTH patients probably have a ‘common’ peripheral inflammatory reaction. Probably different in migraineurs? Maybe, you can give a few comments on this possibility?

Author Response

Nov 1, 2022 Reviewer 3 Journal of Clinical Medicine Dear Reviewer 3, Please find attached a revised version of our manuscript, “Role of Peripheral Inflammatory Markers in Patients with Acute Headache Attack to Differentiate Between Migraine and Non-Migraine Headache” (jcm-2007998). We thank you for your thoughtful suggestions regarding the original version of our paper; most of the suggested changes have been incorporated into the revision. All revisions are described in detail in the order mentioned in the review, following your comment (in italics). We believe that the revisions have greatly improved the manuscript and hereby submit the revised version for your consideration for publication. Comments to author: The purpose of the article was to investigate the role of peripheral inflammatory markers (PIMs) in differentiating migraine from non-migraine headache (nMH) in patients with acute headache attacks. A retrospective study over a ten-year period. The study showed that PIMs were pathologic compared to controls in patients with migraine as well as in patients with nMH. However, the two patient groups have some similarities. We thank the reviewer for these comments and suggestions, which have improved our manuscript. We agree with the reviewer. Our non-migraine headache group included patients with diagnostic codes for TTH or non-specific headache and excluded all types of migraines. Thus, the non-migraine headache group mainly included patients with TTH. Migraine and tension-type headache (TTH) are two most common types of primary headache. Although the International Headache Society had proposed clear criteria for differentiating them, this can be very difficult clinically. Considering the overlapping features of these headaches, a previous study suggested that migraine and TTH are on the same continuum. Due to the retrospective nature of our study, detailed clinical information on headache characteristics was not collected. This is described as a limitation of our study. Absolutely, a well-done study and an interesting one. I have no serious comments to introduction, materials and methods, as well as results. Only, I think it could be valuable that a statistician took an overview for control of PSM. Thank you. As suggested, the statistics were again reviewed by experts, and we have revised the statistics portion of the Materials and Methods as follows: We used logistic regression as the estimation algorithm and the nearest neighbor algorithm as the matching algorithm without replacement. The propensity scores ranged from 0.01 to 0.43 using calipers with a width equal to 0.2 of the standard deviation of the logit. After matching, the absolute standardized mean differences (ASD) for the covariates were < 0.05, indicating adequate balance. (page 4, lines 156 – page 4, lines 161) Discussion: I miss some considerations about PIMs of the nMH group. I assume that most of the nMH patients suffer from tension-type headache (TTH). Probably, TTH suffer from a simple muscular inflammation. In rows 247 – 51: Poor ability of NMR and NLR to differentiate between migraine and nMH, but fair between migraine and control. Thus, migraine patients and nMH patients have similar PIMs findings. However, they do not necessarily have the same cause of the findings. I guess there can be different causes of the PIMs findings. TTH patients probably have a ‘common’ peripheral inflammatory reaction. Probably different in migraineurs? Maybe, you can give a few comments on this possibility? Thank you for the important comments. We have added the following to the Discussion: Systemic pro-inflammatory status is the most robust experimental finding in migraine. However, it is still necessary to understand the extent to which the proinflammatory status of migraineurs reduces the threshold for trigeminovascular system activation, which is believed to promote neuroinflammatory events during migraine attacks [36]. Neuroinflammation has an important role in the pathophysiology of migraine, which is a complex neuro-glio-vascular disorder. Recently, the contribution of activation of the inflammasome, a key component of the innate immune system responsible for multiprotein complex signaling, to the pain signaling system has received attention. Its activation causes the production of inflammatory cytokines that can stimulate trigeminal neurons and are implicated in the generation of migraine pain.[37]. (page 10, lines 287 – page 10, lines 296) Our nMH group included patients with diagnostic codes for TTH or non-specific headache; all types of migraine were excluded. Thus, the nMH group mainly included patients with TTH. (page 10, lines 315 – page 10, lines 317) A previous study showed that knowledge of the different profiles of inflammatory markers can shed light on the pathophysiological differences between migraine and TTH. As different pathophysiological mechanisms have been proposed for migraine and TTH, the molecular mechanisms might also be distinct [39]. Increased tenderness to palpation of the pericranial myofascial tissues, indicative of peripheral pain mechanisms, is the most obvious abnormality in patients with TTH. Central neuroplastic changes may affect the regulation of peripheral mechanisms and lead to, increased pericranial muscle activity or neurotransmitter release in myofascial tissues [40]. Thus, the increase in PIMs in our nMH group, including TTH, is thought to be due to a mechanism different from that in the migraine group. In particular, elevated PIMs in TTH may be associated with a common peripheral inflammatory reaction originating from the myofascial tissue. (page 10, lines 319 – page 10, lines 329) We have also added these citations. 36. Biscetti, L.; Vanna, GD.; Cresta, E.; Bellotti, A.; Corbelli, L.; Cupini, ML.; Calabresi, P.; Sarchielli, P. Immunological findings in patients with migraine and other primary headaches: a narrative review. Clinical and Experimental Immunology, 2022, 207, 11–26. https://doi.org/10.1093/cei/uxab025. 37. Kursun O, Yemisci, M.; Maagdenberg, AM.; Karatas H. Migraine and neuroinflammation: the inflammasome perspec-tive. The Journal of Headache and Pain 2021, 22,55. https://doi.org/10.1186/s10194-021-01271-1. 39. Domingues, RB.; Duarte, H.; Senne, C.; Bruniera, G.; Brunale, F.; Rocha NP.; Teixeira. AL. Serum levels of adiponectin, CCL3/MIP-1α, and CCL5/RANTES discriminate migraine from tension-type headache patients. Arq. Neuro-Psiquiatr. 2016,74(8). https://doi.org/10.1590/0004-282X20160096. 40. Bendtsen, L. Central Sensitization in Tension-Type Headache—Possible Pathophysiological Mechanisms. Cephalal-gia. 2000 Jun,20(5),486-508. doi: 10.1046/j.1468-2982.2000.00070.x We have addressed all of the issues raised by the reviewers. We are grateful for the constructive comments made during the review process. We believe that our paper has been improved by implementing these suggestions. Yours faithfully, Jong-Hee Sohn, M.D. Ph.D. Department of Neurology, Chuncheon Sacred Heart Hospital, Hallym University College of Medicine, 77 Sakju-ro, Chuncheon-si, Gangwon-do, 24253, Republic of Korea Tel: +82-33-252-9970, Fax: +82-33-241-8063 E-mail: deepfoci@hallym.or.kr

Round 2

Reviewer 1 Report

Authors have improved the quality of the paper. Well written revision. I recommend accept for publication.

Kind regards